# SGD with Weight Decay Secretly Minimizes the Ranks of Your Neural Networks

Tomer Galanti[1], Zachary Siegel[2], Aparna Gupte[3], Tomaso Poggio[3]
[1]Texas A&M University, [2]Princeton University, [3]Massachusetts Institute of Technology
galanti@tamu.edu, zss@princeton.edu, agupte@mit.edu, tp@csail.mit.edu

We explore the implicit bias of Stochastic Gradient Descent (SGD) toward learning low-rank weight matrices during the training of deep neural networks. Through theoretical analysis and empirical validation, we demonstrate that this rank-minimizing bias becomes more pronounced with smaller batch sizes, higher learning rates, or stronger weight decay. Unlike previous studies, our analysis does not rely on restrictive assumptions about data, convergence, optimality of the learned weight matrices, network architecture, making it applicable to a wide range of neural network architectures of any width or depth. We further show that weight decay is essential for inducing this low-rank bias. Finally, we empirically explore the connection between this bias and generalization, finding that it has a noticeable, yet marginal, effect on the test performance.

## 1. Introduction

Stochastic gradient descent (SGD) is one of the most widely used optimization techniques for training deep learning models [1]. While initially developed to mitigate the computational challenges of traditional gradient descent, recent studies suggest that SGD also plays a crucial role in regularization, helping prevent overparameterized models from converging to minima that do not generalize well [2–5]. Empirical research has shown, for example, that SGD can outperform gradient descent [5], with smaller batch sizes leading to better generalization [4, 6]. However, the full extent of SGD's regularization effects is not yet fully understood.

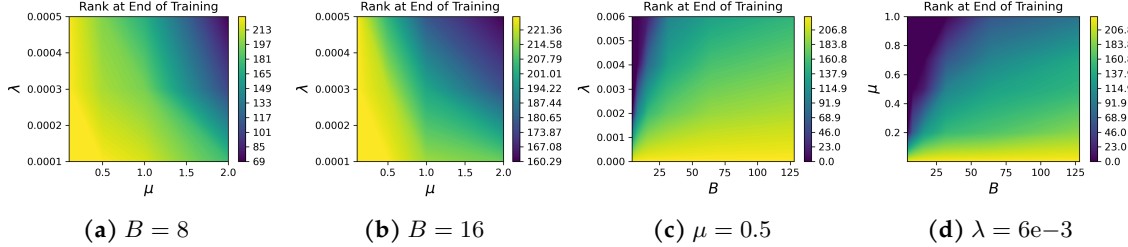

**(a)** $B = 8$     **(b)** $B = 16$     **(c)** $\mu = 0.5$     **(d)** $\lambda = 6\mathrm{e}{-3}$

Figure 1: **Higher weight decay ($\lambda$) and learning rate ($\mu$), or smaller batch sizes ($B$), lead to a lower average rank across the network layers.** We plot the average rank at end of training for ResNet-18 trained on CIFAR10 when varying a pair of hyperparameters.

Various studies have shown that when training large neural networks using gradient-based optimization, the networks tend to become highly compressible. For example, many papers have demonstrated that a significant portion of the weights can be pruned post-training [7–12], large pretrained models can be distilled into smaller models [13–16], and in some cases, the learned weight matrices can be approximated using low-rank matrices without a significant loss in accuracy [17–22]. Beyond compressibility, the inherent low-rank biases of weight matrices has been leveraged for computationally efficient fine-tuning methods, such as LoRA and its variants [23–27], to adapt large pre-trained models at reduced cost. For instance, in [27] they showed that deep overparameterized networks' training dynamics remain confined to low-dimensional subspaces, enabling compressed

Second Conference on Parsimony and Learning (CPAL 2025).

Table 1: **The assumptions and results of various papers on low-rank bias in deep learning.** The last column shows the result of each paper. The notation $\text{Lin}_L$ denotes a composition of $L$ linear layers, and $\sigma$ represents the ReLU activation. N/A is used when the paper does not specify a constraint. Our paper considers a much more realistic setting than all of the previous papers.

| Paper | Architecture | Data | Objective function | Optimizer | Convergence | Result |
|---|---|---|---|---|---|---|
| [31] | $\text{Lin}_L$ | Linearly separable | Exponential/logistic loss | GF | N/A | Each layer has rank $\leq 1$ |
| [33] | $\text{Lin}_1 \circ \sigma \circ \text{Lin}_1$ | 1-dimensional | Min $L_2$ regularization s.t. data fitting | N/A | Global optimum | First layer has rank $\leq 1$ |
| [29] | $\text{Lin}_1 \circ \sigma \circ \text{Lin}_L$ | $d$-dimensional | Min $L_2$ regularization s.t. data fitting | N/A | Global optimum | Bottom linear transformation has rank $\leq d$ |
| [32] | $\text{Lin}_K \circ \sigma \circ \text{Lin}_1 \circ \ldots \circ \sigma \circ \text{Lin}_1$ | Linearly separable | Exponential/logistic loss | GF | N/A | Top $K$ layers have rank $\leq 1$ |
| [30] | $\text{Lin}_1 \circ \sigma \circ \text{Lin}_1 \circ \ldots \circ \sigma \circ \text{Lin}_1$ | Separable by a depth $L'$ network | Min $L_2$ regularization s.t. fitting the data | N/A | Global optimum | Harmonic mean of $\sqrt{\text{stable ranks}} \leq \mathcal{O}(\exp(c/L))$ |
| Ours | Res, Conv, Lin, Activation | N/A | Differentiable loss + $L_2$ regularization | SGD | N/A | Each layer has rank $\mathcal{O}(1)$ (w.r.t the width) |

factorizations that preserve the benefits of overparameterization and yield improved efficiency in both matrix completion and a refined version of LoRA method (Deep LoRA) for LLM adaptation.

While these empirical and recent theoretical results are promising, effectively controlling compression still requires deeper insights into how hyperparameters, data, and architectural choices drive the low-rank bias. To address this gap, several attempts have been made to explore the origins of the low-rank bias. For instance, in [28–30], researchers examined the rank of weight matrices in neural networks that globally minimize $L_2$ regularization while fitting the training data. Specifically, [28] demonstrated that for data on a 1-dimensional manifold, the weight matrices of a two-layer network become rank-1 at the global minimum. This result was later extended in [29], showing that the weight matrix has rank $\leq d$ when the data lies in a $d$-dimensional space. Additionally, [30] found that in sufficiently deep ReLU networks (of depth $L$), the harmonic mean of the square roots of the stable ranks of the weight matrices decays at a rate of $\exp(c/L)$ at the global minimum of $L_2$ regularization, subject to fitting the training data. Similarly, [31] showed that gradient flow (GF) training of univariate linear networks with exponentially-tailed classification losses learns rank-1 weight matrices when the data is linearly separable. A more recent study [32] extended this result, demonstrating that when training a ReLU network with multiple linear layers at the top using GF, the top layers converge to rank-1 weight matrices.

While these analyses offer valuable insights, each one makes strong assumptions about the structure of the data (e.g., linear separability, low-dimensionality), the network architecture, the optimization method, or the objective function, and they only apply to specific layers of the network. Moreover, these analyses reveal little about the relationship between the low-rank bias and the training hyperparameters or the model architecture, which limits the practical utility of these results.

**Contributions.** In this paper, we show that using mini-batch Stochastic Gradient Descent (SGD) and weight decay *implicitly minimizes the rank* of the learned weight matrices during the training of neural networks, particularly encouraging the *learning of low-dimensional feature manifolds*. Within the active field that investigates these properties [30–34], our analysis is the first to characterize how *SGD and weight decay induce a low-rank bias in all of the weight matrices of a wide range of neural network architectures* (e.g., with residual connections [35], self-attention layers [36] and convolutional layers [37]), without making assumptions about the data (e.g., linear separability or low-dimensionality) or strong assumptions about the convergence of the training process. Our theoretical analysis predicts that *smaller batch sizes, higher learning rates, or increased weight decay results in a decrease in the rank of the learned matrices, and that weight decay is necessary for this bias to emerge*. Our results are compared with the previous literature in Table 1.

To validate our theory, we provide a comprehensive empirical analysis in which we examine the regularization effects of different hyperparameters on the rank of weight matrices for various network architectures. Additionally, we carried out several experiments to examine the connection between low-rank bias and generalization. Our findings suggest that although low-rank bias is not crucial for good generalization, it is correlated with a slight improvement in performance.

## 2. Problem Setup

We study the influence of using mini-batch SGD in conjunction with weight decay on the ranks of the learned weight matrices of neural networks in standard learning settings. Namely, we consider a parametric model $\mathcal{F} \subset \{f' : \mathcal{X} \to \mathbb{R}^q\}$, where each function $f_W \in \mathcal{F}$ is specified by a vector of parameters $W \in \mathbb{R}^N$. The goal is to learn a function from a training dataset $S = \{x_i\}_{i=1}^m$. For each sample we have a loss function measuring the performance on that sample $\ell_i : \mathbb{R}^q \to \mathbb{R}$ which is simply a differentiable function. For example, in supervised learning we have $\ell_i(u, y_i)$, where $y_i$ is the label of the $i$th sample. The model is trained to minimize the regularized empirical risk,

$$L_S^\lambda(f_W) := \frac{1}{m} \sum_{i=1}^m \ell_i(f_W(x_i)) + \lambda \|W\|_F^2, \tag{1}$$

where $\lambda > 0$ is a predefined hyperparameter and $\| \cdot \|_F$ is the Frobenius norm. To accomplish this task, we typically use mini-batch SGD, as outlined in the following paragraph.

**Model architecture.**  In this paper, we consider a broad set of neural network architectures, including but not limited to neural networks with fully-connected layers, residual connections, convolutional layers, pooling layers, sub-differentiable activation functions (e.g., sigmoid, tanh, ReLU, Leaky ReLU), self-attention layers and siamese layers.

In this framework, the model $f_W(x) := h(x; W^1, \ldots, W^k)$ is a function that takes a sequence of weight matrices $W^1, \ldots, W^k$ and an input vector $x$. Throughout the paper, we assume that for each layer $l \in [k]$, we can write

$$f_W(x) = g_l(W^l u_1^l(x, W_{|l}), \ldots, W^l u_{m_l}^l(x, W_{|l}), W_{|l}, x), \tag{2}$$

where $g_l$ is a sub-differentiable function accepting vectors $W^l u_j^l(x, W_{|l})$, the parameters $W_{|l} = \{W^j\}_{j \neq l}$ and $x$ as input. Here, $u_j^l(x, W_{|l})$ are functions of $x$ and the weight matrices $W_{|l}$, viewed as a layer preceding $W^l$.

For example, a neural network with a fully-connected layer can be written as follows:

$$f_W(x) = g_l(W^l u^l(x, W_{|l}), W_{|l}, x), \tag{3}$$

where $u^l(x, W_{|l})$ is the input to the fully-connected layer and $g_l(z, W_{|l}, x)$ takes the output $z = W^l u^l(x, W_{|l})$ of the fully-connected layer and returns the output of the neural network (e.g., by composing multiple layers on top of it). More specifically, a fully-connected network $f_W(x) = W^L \sigma(W^{L-1} \cdots W^2 \sigma(W^1 x) \cdots)$ can be written as $g_l(W^l u^l(x, W_{|l}))$, where $g_l(z) = W^L \sigma(W^{L-1} \cdots W^{l+1} \sigma(z) \cdots)$ and $u^l(x, W_{|l}) = \sigma(W^{l-1} \cdots W^2 \sigma(W^1 x) \cdots)$. We can also represent convolutional layers within this framework. We can think of a convolutional layer as a transformation that takes some input $u$ and applies the same linear transformation to multiple 'patches' independently, $W^l u_1^l, \ldots, W^l u_{m_l}^l$, where each $u_j^l$ denotes a patch in the input $u$ (a patch in this case is a subset of the coordinates in $u$), $m_l$ is the number of patches and $W^l \in \mathbb{R}^{c_l \times c_{l-1} d_{l-1}}$ is a matrix representation of the kernel. In this case, $u^l(x, W_{|l})$ is the output of the layer below the convolutional layer and $u_j^l(x, W_{|l})$ is the $j$th patch that the convolutional layer is applied to. Furthermore, $g_l$ is simple the composition of the layers following the given convolutional layer. In similar ways, we can also express neural networks with residual connections, self-attention layers, hypernetwork layers, pooling layers, etc'.

**Optimization.**  We employ stochastic sub-gradient descent (SGD) to minimize the regularized empirical risk $L_S^\lambda(f_W)$ over a specified number of iterations $T$. We begin by initializing $W_1$ at some point. We split the data $S$ into $r = \frac{|S|}{B}$ batches of size $B$ (for simplicity we assume that $|S|$ is divisible by $B$) and at iteration $t$, we take batch $\tilde{S}_{t'}$ with $t' = (t \mod r)$ and update $W_{t+1} = W_t - \mu \nabla_W L_{\tilde{S}_{t'}}^\lambda(f_{W_t})$, where $\mu > 0$ is the predefined learning rate and $\nabla_W g(W)$ represents a sub-gradient of $g : \mathbb{R}^n \to \mathbb{R}$. We use sub-gradient descent when dealing with models that are only sub-differentiable, such as ReLU neural networks (for more details, see section 14.2 in [38]). It is worth noting that when the model is differentiable, sub-gradient descent aligns with gradient descent.

# 3. Theoretical Results

In this section, we prove that when training neural networks with regularized SGD for a long time, the weight matrices can be approximated with matrices of bounded ranks. We begin by making a simple observation that the rank of $\nabla_{W^l}\ell(f_W(x))$ is bounded by $m_l$ (see 'Model architecture' in section 2) for any $l$ and any sample $x$. Then, by recursively unrolling the optimization process, we express the weight matrix $W_T^l$ as a sum of $(1 - \mu\lambda)^n W_{T-n}^l$ and $nB$ gradients of the loss function with respect to $W^l$ for different samples at different iterations. Since each one of these terms is a matrix of rank $\leq m_l$, we conclude that the distance between $W_T^l$ and a matrix of rank $\leq m_l Bn$ decays exponentially fast when increasing $n$.

> **Lemma 3.1.** *Let $\ell$ be a differentiable loss function, and let $f_W$ be a model as described in section 2. For any weight matrix $W^l$ in $f_W$ and any sample $x \in \mathbb{R}^d$, the following inequality holds:*
>
> $$\mathrm{rank}\left(\nabla_{W^l}\ell(f_W(x))\right) \leq m_l,$$
>
> *where $m_l$ is a constant depending on the structure of the layer $l$ (defined in Equation 2).*

*Proof.* Let $u_j = u_j^l(x, W_{|l})$, and let $f_W(x)_r$ denote the $r$-th coordinate of $f_W(x)$. By applying the chain rule, we have the following identity for the gradient:

$$\nabla_{W^l}\ell(f_W(x)) = \sum_{r=1}^q \frac{\partial\ell(f_W(x))}{\partial f_W(x)_r} \cdot \frac{\partial f_W(x)_r}{\partial W^l}.$$

Next, observe that:

$$\frac{\partial f_W(x)_r}{\partial W^l} = \sum_{j=1}^{m_l} \frac{\partial f_W(x)_r}{\partial W^l u_j} \cdot \frac{\partial W^l u_j}{\partial W^l} = \sum_{j=1}^{m_l} \frac{\partial f_W(x)_r}{\partial W^l u_j} \cdot u_j^\top.$$

Substituting this into the previous expression and reordering the sums, we get:

$$\nabla_{W^l}\ell(f_W(x)) = \sum_{r=1}^q \frac{\partial\ell(f_W(x))}{\partial f_W(x)_r} \cdot \sum_{j=1}^{m_l} \frac{\partial f_W(x)_r}{\partial W^l u_j} \cdot u_j^\top = \nabla_{W^l}\ell(f_W(x)) = \sum_{j=1}^{m_l} \left( \sum_{r=1}^q \frac{\partial\ell(f_W(x))}{\partial f_W(x)_r} \cdot \frac{\partial f_W(x)_r}{\partial W^l u_j} \right) u_j^\top.$$

This represents a sum of $m_l$ outer products of vectors, implying that the resulting matrix has a rank of at most $m_l$. $\square$

The above lemma shows the rank of the gradient with respect to any weight matrix $W^l$ is bounded by $\leq m_l$. In particular, for a fully-connected layer with weight matrix $W^l$, the sub-gradient of the loss function with respect to $W^l$ is at most 1 and for a convolutional layer it is bounded by the number of patches upon which the kernel is being applied.

The following lemma provides an upper bound on the minimal distance between the network's weight matrices and matrices of bounded rank.

> **Lemma 3.2.** *Let $\|\cdot\|$ be any matrix norm and $\ell$ any differentiable loss function. Consider a model $f_W$ as described in section 2 and $W^l$ be a weight matrix within $f_W$. Suppose we train $f_W$ using SGD with batch size $B \in [m]$, learning rate $\mu > 0$ and weight decay $\lambda > 0$, where $m$ is the total number of training samples. Then, for any integer $T > n$, the following inequality holds:*
>
> $$\min_{\bar{W}^l:\ \mathrm{rank}(\bar{W}^l) \leq m_l Bn} \left\| \frac{W_T^l}{\|W_T^l\|} - \bar{W}^l \right\| \leq (1 - 2\mu\lambda)^n \cdot \frac{\|W_{T-n}^l\|}{\|W_T^l\|}.$$

*Proof.* Let $\tilde{S}_t \subset S$ the mini-batch that was used by SGD at iteration $t$. We have

$$\begin{aligned} W_T^l &= W_{T-1}^l - \mu\nabla_{W^l}L_{\tilde{S}_{T-1}}(f_{W_{T-1}}) - 2\mu\lambda W_{T-1}^l \\ &= (1 - 2\mu\lambda)W_{T-1}^l - \mu\nabla_{W^l}L_{\tilde{S}_{T-1}}(f_{W_{T-1}}). \end{aligned}$$

Similarly, we can write

$$W^l_{T-1} = (1 - 2\mu\lambda)W^l_{T-2} - \mu\nabla_{W^l}L_{\tilde{S}_{T-2}}(f_{W_{T-2}}).$$

This gives us

$$W^l_T = (1 - 2\mu\lambda)^2 W^l_{T-2} - \mu\nabla_{W^l}L_{\tilde{S}_{T-1}}(f_{W_{T-1}}) - \mu(1 - 2\mu\lambda)\nabla_{W^l}L_{\tilde{S}_{T-2}}(f_{W_{T-2}}).$$

By recursively applying this process $n$ times, we have

$$W^l_T = (1 - 2\mu\lambda)^n W^l_{T-n} - \mu\sum_{j=1}^{n}(1 - 2\mu\lambda)^{j-1}\nabla_{W^l}L_{\tilde{S}_{T-j}}(f_{W_{T-j}}) =: (1 - 2\mu\lambda)^n W^l_{T-n} + U^l_{T,n}.$$

We notice that,

$$\nabla_{W^l}L_{\tilde{S}_{T-j}}(f_{W_{T-j}}) = \frac{1}{B}\sum_{x_i \in \tilde{S}_{T-j}}\nabla_{W^l}\ell_i(f_{W_{T-j}}(x_i)).$$

According to Lemma 3.1, we have $\mathrm{rank}(\nabla_{W^l}\ell_i(f_{W_{T-j}}(x_i))) \leq m_l$. Therefore, $\mathrm{rank}(\nabla_{W^l}L_{\tilde{S}_{T-j}}(f_{W_{T-j}})) \leq Bm_l$ since $\nabla_{W^l}L_{\tilde{S}_{T-j}}(f_{W_{T-j}})$ is an average of $B$ matrices of rank at most $m_l$. In particular, $\mathrm{rank}(U^l_{T,n}) \leq m_l Bn$ since $U^l_{T,n}$ is a sum of $n$ matrices of rank at most $m_l B$. Therefore, we obtain that

$$\min_{\bar{W}^l:\ \mathrm{rank}(\bar{W}^l)\leq m_l Bn}\left\|W^l_T - \bar{W}^l\right\| \leq \left\|W^l_T - U^l_{T,n}\right\| = (1 - 2\mu\lambda)^n\|W^l_{T-n}\|.$$

Finally, by dividing both sides by $\|W^l_T\|$ we obtain the desired inequality. $\square$

The lemma above provides an upper bound on the minimal distance between the parameters matrix $W^{ij}_T$ and a matrix of rank $\leq m_l Bn$. The parameter $t$ is a parameter of our choice that controls the looseness of the bound and is independent of the optimization process. The bound is proportional to $(1 - 2\mu\lambda)^n \frac{\|W^l_{T-n}\|}{\|W^{ij}_T\|}$, which decreases exponentially with $n$ as long as $\|W^l_T\|$ converges to a non-zero value. As a next step, we tune $t$ to ensure that the bound would be smaller than $\epsilon$. This result is formalized in the following theorem.

> **Theorem 3.3.** *Let $\|\cdot\|$ be any matrix norm and $\ell$ a differentiable loss function and $\mu, \lambda > 0$ such that $\mu\lambda < 0.5$, $B \in [m]$, and $\epsilon > 0$. Consider a model $f_W$ as described in section 2 and $W^l$ be a weight matrix within $f_W$. Suppose we train $f_W$ using SGD with batch size $B \in [m]$, learning rate $\mu > 0$ and weight decay $\lambda > 0$, where $m$ is the total number of training samples. Assume that $\lim_{T\to\infty}(\|W^l_{T-1}\|/\|W^l_T\|) = 1$. Then, for sufficiently large $T$,*
>
> $$\min_{\bar{W}^l:\ \mathrm{rank}(\bar{W}^l)\leq \frac{m_l B\log(2/\epsilon)}{2\mu\lambda}}\left\|\frac{W^l_T}{\|W^l_T\|} - \bar{W}^l\right\| \leq \epsilon.$$

*Proof.* We pick $n = \lceil\frac{\log(\epsilon/2)}{\log(1-2\mu\lambda)}\rceil$. Since $n$ is independent of $T$, we have

$$\lim_{T\to\infty}\frac{\|W^l_{T-n}\|}{\|W^l_T\|} = \lim_{T\to\infty}\prod_{j=1}^{n}\frac{\|W^l_{T-j}\|}{\|W^l_{T-j+1}\|} = \prod_{j=1}^{n}\lim_{T\to\infty}\frac{\|W^l_{T-i}\|}{\|W^l_{T-i+1}\|} = 1.$$

Then, for any sufficiently large $T$, we have $\frac{\|W^l_{T-n}\|}{\|W^l_T\|} \leq 2$.

We notice that for the selected $T$, we have $(1 - \mu\lambda)^n \leq \epsilon/2$. Hence, for any large $T$, we have,

$$(1 - \mu\lambda)^n\frac{\|W^l_{T-n}\|}{\|W^l_T\|} \leq \epsilon$$

Furthermore, since $\mu\lambda < 0.5$, we also have $n \leq \frac{\log(2/\epsilon)}{2\mu\lambda}$ and $m_l Bn \leq \frac{m_l B\log(2/\epsilon)}{2\mu\lambda}$. Therefore, by Lemma 3.2, we have the desired inequality. $\square$

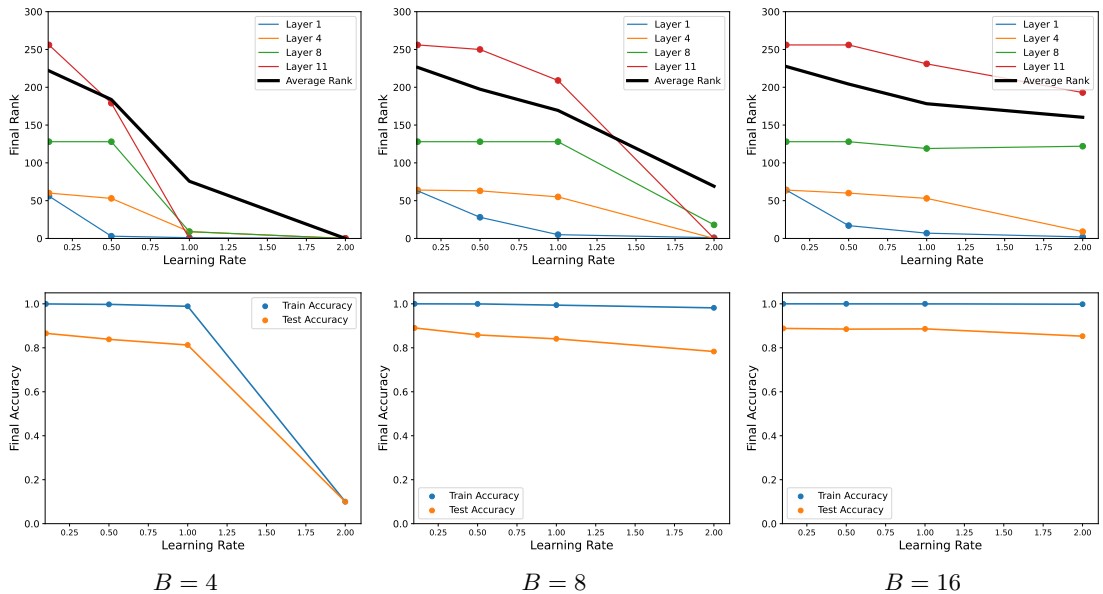

Figure 2: **Average ranks and accuracy rates of ResNet-18 trained on CIFAR10 when varying $\mu$.** The top row shows the average rank across layers, while the bottom row shows the train and test accuracy rates for each setting. In this experiment, $\lambda = 5e{-}4$ and $\epsilon = 1e{-}3$.

The above theorem provides an upper bound on the rank of the learned weight matrices. It shows that when training the model, the normalized weight matrices $\frac{W_T^l}{\|W_T^l\|}$ become approximately matrices of rank at most $\frac{m_l B \log(2/\epsilon)}{2\mu\lambda}$. While $\frac{m_l B \log(2/\epsilon)}{2\mu\lambda}$ is not necessarily a small number, this bound is still nontrivial since $\frac{m_l B \log(2/\epsilon)}{2\mu\lambda} = \mathcal{O}(1)$ with respect to the iteration $t$ and the size of the network (its width, depth, etc'). This result is particularly striking as it reveals a mechanism that encourages learning low-rank weight matrices that exclusively depends on the optimization process of SGD with weight decay, regardless of the weight initialization, geometric properties of the data, or dimensionality of the data, which are largely irrelevant to the analysis. The assumption $\lim_{T \to \infty} (\|W_{T-1}^l\|/\|W_T^l\|) = 1$ generally occurs in practice and is validated in Appendix B. As a special case, it holds when $\|W_T^l\|$ converges to a non-zero value.

## 4. Experiments

In the previous section, we have seen that one can approximate the learned weight matrices using matrices of bounded rank. Since the bound becomes smaller as we increase $\lambda$, $\mu$, or decrease $B$, we make the following prediction:

**Prediction 4.1.** *When training a neural network using SGD with weight decay, the effective rank of the learned weight matrices tends to decrease as the batch size decreases, or as the weight decay or learning rate increases.*

Although the effective rank of a given matrix can be measured in various ways, in the experiments we will focus on counting how many singular values of the normalized version of the matrix exceed a predefined threshold $\epsilon > 0$ (we use $\epsilon = 1e{-}3$ unless otherwise stated). In order to validate this prediction, we empirically study how batch size, weight decay, and learning rate affect the rank of matrices in deep networks. We conduct separate experiments in which we vary one hyperparameter while keeping the others constant to isolate its effect on the average rank. Additional experiments with a variety of architectures (e.g. ViT, ResNet-18 and VGG-16), data sets (e.g., CIFAR10, MNIST,

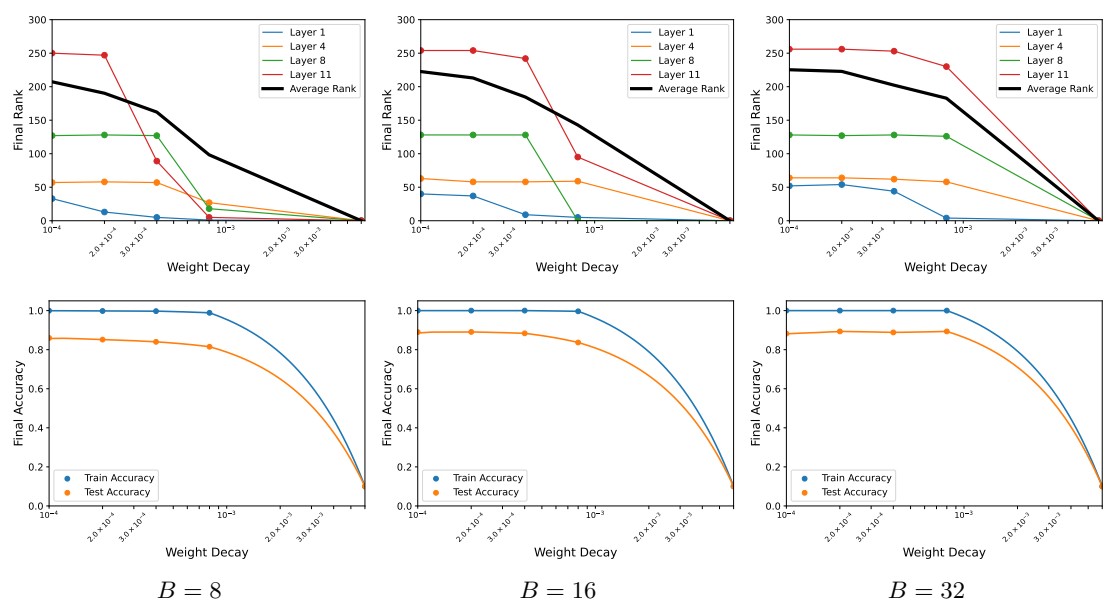

Figure 3: **Average ranks and accuracy rates of ResNet-18 trained on CIFAR10 when varying** $\lambda$. In this experiment, $\mu = 1.5$ and $\epsilon = 1\mathrm{e}{-3}$.

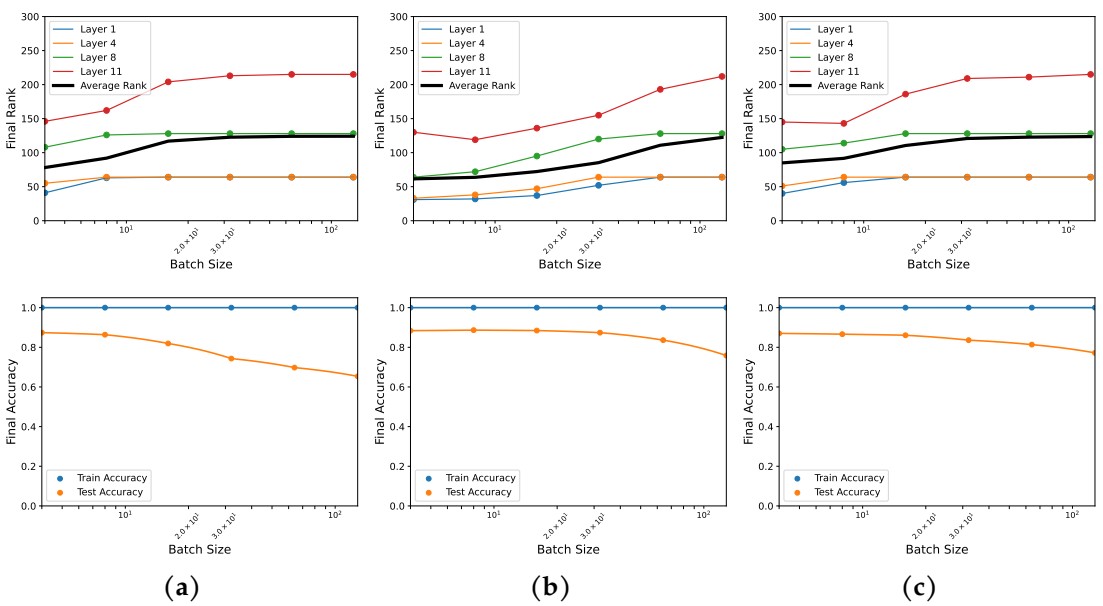

Figure 4: **Average ranks and accuracy rates of ResNet-18 trained on CIFAR10 when varying** $B$. In (**a**) we used $\mu = 1\mathrm{e}{-3}$ and $\lambda = 6\mathrm{e}{-3}$, in (**b**) we used $\mu = 5\mathrm{e}{-3}$ and $\lambda = 6\mathrm{e}{-3}$, and in (**c**) we used $\mu = 1\mathrm{e}{-2}$ and $\lambda = 4\mathrm{e}{-4}$. We used a threshold of $\epsilon = 1\mathrm{e}{-3}$.

Fashion MNIST, SVHN, Food101 and Imagenette), and visualizations of singular values of the weight matrices are provided in Appendix B. The plots are best viewed when zooming in on the pictures. Each of the runs was done using a single GPU for at most 60 hours on a computing cluster with several available GPU types (e.g., GeForce RTX 2080, Tesla V-100).

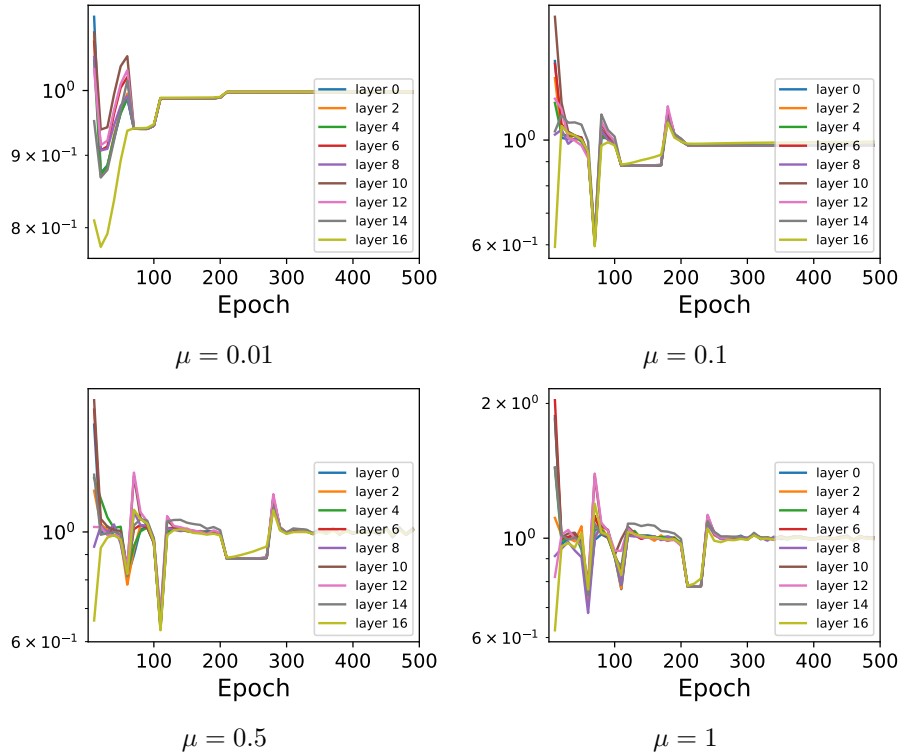

Figure 5: **Convergence of the weights for ResNet-18 trained on CIFAR10.** In this experiment, $B = 8$, $\lambda = 5\mathrm{e}{-4}$ and $\epsilon = 0.01$ (see Figure 2(mid) in the main text for the weight ranks and accuracy rates).

## 4.1. Setup

**Architectures.** We consider four types of network architectures. (**i**) The first architecture is a multi-layer perceptron (MLP), denoted as MLP-BN-$L$-$H$, which comprises $L$ hidden layers, each containing a fully-connected layer with width $H$, followed by batch normalization and ReLU activations. This architecture ends with a fully-connected output layer. (**ii**) The second architecture is the convolutional network (VGG-16) proposed by [39], with dropout replaced by batch normalization layers to improve training performance, and a single fully-connected layer at the end. (**iv**) The fourth architecture is the residual network (ResNet-18) proposed in [35]. (**v**) The fifth architecture is a small visual transformer (ViT) [40]. We used a standard ViT that splits the input images into patches of size $4 \times 4$, and includes $8$ self-attention heads, each composed of $6$ self-attention layers. The self-attention layers are followed by two fully-connected layers with a dropout probability of $0.1$, and a GELU activation in between them.

**Training.** To study the hyperparameters' influence on the rank of the weight matrices, we trained models while varying one hyperparameter at a time, while keeping other hyperparameters constant. We trained each model for classification using Cross-Entropy loss minimization between its logits and the one-hot encodings of the labels. The training was carried out by SGD with batch size $B$, initial learning rate $\mu$, and weight decay $\lambda$. The MLP-BN-$L$-$H$, ResNet-18, and VGG-16 models were trained with a decreasing learning rate of $0.1$ at epochs $60$, $100$, and $200$, and the training was stopped after $500$ epochs. The ViT models were trained using SGD with a learning rate that was decreased by a factor of $0.2$ at epochs $60$ and $100$ and training was stopped after $200$ epochs. During training, we applied random cropping, random horizontal flips, and random rotations (by $15k$ degrees for $k$ uniformly sampled from [24]) and standardized the data.

**Evaluation of the rank.**   After each epoch, we compute the average rank across the network's weight matrices and its train and test accuracy rates. For a convolutional layer, we represent its kernel parameters as a matrix, whose rows are vectorized versions of its kernels.

To estimate the rank of a given matrix $W$, we count how many of the singular values of $\frac{W}{\|W\|_2}$ are greater than $\epsilon$ (namely, $\# \left\{ i \mid \sigma_i \left( \frac{W}{\|W\|_2} \right) > \epsilon \right\}$), where $\epsilon$ is a small tolerance value (we use $\epsilon = 1\mathrm{e}{-3}$ by default). We note that the number of singular values greater than $\epsilon$ is closely related to the bound described in Theorem 3.3. Namely, by the Eckart-Young-Mirsky theorem, we have:

$$r = \# \left\{ i \mid \sigma_i \left( \frac{W}{\|W\|_2} \right) > \epsilon \right\} \quad \Longleftrightarrow \quad \min_{r \in \mathbb{N}} \min_{\bar{W}:\mathrm{rank}(\bar{W}) \leq r} \left\| \frac{W}{\|W\|_2} - \bar{W} \right\|_2 \leq \epsilon,$$

where $\sigma_i(A)$ is the $i$th singular value of the matrix $A$.

## 4.2. Results

**Validating prediction 4.1.**   As shown in Figures 1-4, a smaller batch size or higher learning rate and weight decay leads to a smaller effective ranks across the network layers. These results align with prediction 4.1. For additional validation of this prediction, see all of the experiments in Appendix B.

**Verifying that** $\lim_{T \to \infty} (\|W^l_{T-1}\|/\|W^l_T\|) = 1$**.**   In Theorem 3.3 we made the assumption that $\lim_{T \to \infty} (\|W^l_{T-1}\|/\|W^l_T\|) = 1$. In order to validate this assumption, we trained various models and monitored the ratio between the norms of each layer at consecutive epochs. In each Figure 5 we report the ratios across different layers for a neural network with a certain learning rate. As can be seen, the ratios consistently converge to 1 during training. For a similar experiment with VGG-16 [39] see Figure 17 in Appendix B.

**Low-rank bias and generalization.**   We investigated the relationship between low-rank bias and generalization by training ResNet-18 models on CIFAR10 with varying batch sizes, while keeping $\lambda$ and $\mu$ constant. To provide a fair comparison, we selected $\lambda$ and $\mu$ to ensure all models perfectly fit the training data. Our results, shown in Figure 4, indicate that models trained with smaller batch sizes (and as a result with matrices of lower ranks) tend to have a better test performance. Based on these findings, we predict that when altering a certain hyperparameter, a neural network with a lower average rank will have better test performance than a network with the same architecture but higher rank matrices, assuming both networks perfectly fit the training data. For a similar experiment with VGG-16 [39] see Figure 13 in Appendix B.

# 5. Conclusions

Mathematically characterizing the inductive biases of neural networks trained with SGD remains a significant open problem in the theory of deep learning [41]. In this work, we address one of the key inductive biases observed in empirical studies: the implicit minimization of the rank of learned weight matrices during training. Through our theoretical analysis of the training dynamics of regularized SGD, we identify a forgetting mechanism, where past updates are forgotten exponentially fast, resulting in learned weights that can be approximated by a mixture of recent training updates. This process leads to a rank minimization mechanism influenced by batch size, learning rate, and weight decay. Notably, this behavior appears largely independent of the geometry of the training data or its intrinsic dimensionality.

A promising direction for future work is to explore whether this theoretical framework can shed light on other empirical phenomena, such as emergent sparsity in neural networks [10], Neural Collapse in intermediate layers [42, 43], and Grokking [44]. Additionally, it would be valuable to investigate how other factors, such as momentum, affect the rank of the learned matrices and whether our findings can inspire new algorithms for compressing neural networks. Lastly, it would be interesting to study the relationship between this inductive bias and generalization, which seems plausible based on our empirical observations.

# Acknowledgements

This work was supported by the Center for Brains, Minds and Machines (CBMM), funded by NSF STC award CCF - 1231216.

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

# A. Analyzing the Gradient Rank for Self-Attention Layers

We begin by defining a multi-head self-attention layer with $h$ heads, embedding dimension $d_{\text{model}} = hd_k$, per head key dimension $d_k$, and sequence length $T$. Denote

$$
\begin{aligned}
W_Q &= [W_{Q_1}, ..., W_{Q_h}] \in \mathbb{R}^{d_{\text{model}} \times d_{\text{model}}} \\
W_K &= [W_{K_1}, ..., W_{K_h}] \in \mathbb{R}^{d_{\text{model}} \times d_{\text{model}}} \\
W_V &= [W_{V_1}, ..., W_{V_h}] \in \mathbb{R}^{d_{\text{model}} \times d_{\text{model}}}
\end{aligned}
\tag{4}
$$

In multi-head self-attention, the input matrix $Z \in \mathbb{R}^{T \times d_{\text{model}}}$ is projected into keys, queries, and values for each head: $K_i = ZW_{K_i}$, $Q_i = ZW_{Q_i}$ and $V_i = ZW_{V_i}$. The output of the layer is then

$$
\text{MHA}(Z) = \left[ \text{softmax}\left( \frac{Q_1 K_1^\top}{\sqrt{d_k}} \right) V_1, \ldots, \text{softmax}\left( \frac{Q_h K_h^\top}{\sqrt{d_k}} \right) V_h \right] \in \mathbb{R}^{T \times d_{\text{model}}}
\tag{5}
$$

where the softmax is applied row-wise.

A typical architecture whose $l$th layer is a self-attention layer can be written as follows $f_W(X) = g(u^l(X)W_Q^l, u(X)W_K^l, u^l(X)W_V^l, X, W_{|l})$, where $u^l : \mathbb{R}^{T \times d_{\text{in}}} \to \mathbb{R}^{T \times d_{\text{model}}}$ is a sub-differentiable function that computes the input matrix $Z = u^l(X)$ to the self-attention layer, $W^l = (W_Q^l, W_K^l, W_V^l)$ are the parameters of the $l$th layer, $W_{|l}$ contains all of the model's trainable parameters excluding $W^l$ and $g$ is a sub-differentiable function (the composition of the layers above the $l$th layer).

The reason why we can represent common architectures this way, is because the MHA layer can be written as a differentiable function applied to the $u^l(X)W_Q^l, u(X)W_K^l, u^l(X)W_V^l$ and the layers $g$ on top of the MHA are using sub-differentiable functions using the MHA's output and residual connections. For example, $g$ may include the full self-attention block, which is computed as follows:

$$
\begin{aligned}
Z_1 &= \text{LayerNorm}(\text{MHA}(Z)W_O^{l+1} + u^l(X)), \quad W_O^{l+1} \in \mathbb{R}^{d_{\text{model}} \times d_{\text{model}}} \\
Z_2 &= \text{ReLU}(Z_1 W_1^{l+1} + 1_T \cdot (b_1^{l+1})^\top), \quad W_1^{l+1} \in \mathbb{R}^{d_{\text{model}} \times d}, \ b_1^{l+1} \in \mathbb{R}^d \\
Z_3 &= Z_2 W_2^{l+1} + 1_T \cdot (b_2^{l+1})^\top, \quad b_2^{l+1} \in \mathbb{R}^d \\
\text{Output} &= \text{LayerNorm}(Z_1 + Z_3).
\end{aligned}
$$

where $1_T$ is a vector of 1s of length $T$. The reason why such a function $g$ can implement this block is because the block can be written as $B(Z, W_O^{l+1}, W_1^{l+1}, W_2^{l+1}, b_1^{l+1}, b_2^{l+1}, u^l(X))$, where $W_O^{l+1}, W_1^{l+1}, W_2^{l+1}, b_1^{l+1}, b_2^{l+1}$ are parameters within $W_{|l}$ independent of the parameters of the self-attention layer, $u^l(X)$ can be computed using $X$ and $W_{|l}$ and all of the involved functions are sub-differentiable.

**Lemma A.1.** *Let $\ell$ be a differentiable loss function, $f_W(X) = g(u^l(X)W_Q^l, u^l(X)W_K^l, u^l(X)W_V^l, X, W_{|l})$, where $u^l : \mathbb{R}^{T \times d_{\text{in}}} \to \mathbb{R}^{T \times d_{\text{model}}}$ is a sub-differentiable function that computes the input matrix to the self-attention layer, $W^l = (W_Q^l, W_K^l, W_V^l)$ are the parameters of the lth layer, $W_{|l}$ contains all of the model's trainable parameters excluding $W^l$ and $g$ is a sub-differentiable function (the composition of the layers above the lth layer). Then,*

$$
\text{rank}\left( \nabla_{W_Q} \ell(f_W(X)) \right), \text{rank}\left( \nabla_{W_K} \ell(f_W(X)) \right), \text{rank}\left( \nabla_{W_V} \ell(f_W(X)) \right) \leq T.
$$

*Proof.* Let $X \in \mathbb{R}^{T \times d_{\text{in}}}$ be the input to the network (of sequence length $T$), and let $Z = u^l(X) \in \mathbb{R}^{T \times d_{\text{model}}}$, where $u^l$ is the composition of all layers below layer $l$. By construction, $u^l$ does not depend on $W^l$. By the chain rule:

$$
\begin{aligned}
\frac{\partial \ell(f_W(X))}{\partial W_Q^l} &= \frac{\partial \ell(g(u^l(X)W_Q^l, u(X)W_K^l, u^l(X)W_V^l, X, W_{|l}))}{\partial W_Q^l} \\
&= \frac{\partial \ell(g(u^l(X)W_Q^l, u(X)W_K^l, u^l(X)W_V^l, X, W_{|l}))}{\partial u^l(X)W_Q^l} \cdot \frac{\partial u^l(X)W_Q^l}{\partial W_Q^l} \\
&= \frac{\partial \ell(g(u^l(X)W_Q^l, u(X)W_K^l, u^l(X)W_V^l, X, W_{|l}))}{\partial u^l(X)W_Q^l} \cdot u^l(X).
\end{aligned}
\tag{6}
$$

Since $u^l(X) \in \mathbb{R}^{T \times d_{\text{model}}}$, its rank is at most $T$. Since the rank of a product of two matrices is bounded by the minimal rank of the two matrices, the rank of $\frac{\partial \ell(f_W(X))}{\partial W_Q^l}$ is bounded by $T$. The same argument applies to $W_K$ and $W_V$. □

# B. Additional Experiments

We conducted additional experiments with various learning settings, including training on different datasets and using different architectures, to provide additional evidence for the bias of SGD with weight decay toward rank minimization. The experimental setup and results are described below.

## B.1. Results

**Comparing our bound with the averaged rank.** As mentioned in the main text, our bound $m_l B \log(2/\epsilon)/(2\mu\lambda)$ is generally loose, but not trivial, as it scales as $\mathcal{O}(1)$ relative to the actual dimensions of the weight matrices. To demonstrate that our bound is non-trivial for wide neural networks, we trained an MLP-BN-2-10000 on CIFAR10 using $B = 6$, $\mu = 0.1$, and $\lambda = 8\text{e}{-}3$. As shown in Figure 6, the network is able to train (achieves a non-trivial training accuracy), and at the same time, the bound is strictly smaller than the width of 10000 for any $\epsilon \geq 0.3$.

**Training with different architectures.** In the main text, we validated our predictions using the ResNet-18 architecture. For a more comprehensive analysis, we conducted similar experiments with additional architectures. Similar to the results in the main text, Figures 7, 8,9, 11, 10, 14 and 15 show that, as we increase weight decay or learning rate or decrease batch size, the effective rank of the learned weight matrices tends to decrease.

**Rank minimization bias during training.** To complement our results, we trained ViT models and plotted the ranks during training in Figures 10 and 11. As shown, the training proceeds in two phases: in the first, the rank decreases monotonically, and in the second, it becomes relatively stable and does not change much.

**Training with momentum.** To ensure that our observations are applicable beyond just SGD with weight decay, we conducted an experiment to test whether they also hold for SGD with both weight decay and momentum. As shown in Figure 7, our predictions regarding the regularization effects of hyperparameters remain consistent, even when momentum is included in the training process.

**Training on different datasets.** In Figures 14-15, we trained ResNet-18 instances on the SVHN, Food101, Imagenette, MNIST, Fashion MNIST and Places365 datasets, varying the learning rate while keeping the batch size ($B = 16$) and weight decay ($\lambda = 5\text{e}{-}4$) constant. Since the training on Places365 computationally heavy, we only used 100k samples for training. The observed behavior, previously noted for CIFAR-10, is also replicated for these different datasets.

**Validating the role of positive weight decay for enabling the low-rank bias.** Our theory demonstrates that the low-rank bias emerges when training a model with SGD combined with weight decay. This raises the question of whether weight decay is necessary in practice to achieve the low-rank bias. In Figures 7, 8, 11, and 12, we observe that when $\lambda = 0$, the influence of the batch size or learning rate on the rank of the weight matrices is minimal.

**Comparing the rank of convolutional and fully-connected layers.** In Theorem 3.3, the bound on the rank of the weight matrix of the $l$th layer scales with $m_l$. For a convolutional layer, $m_l$ equals the number of patches on which the convolutional kernel is applied. In contrast, for a fully connected linear layer, $m_l = 1$. This raises the question of whether the rank of the parameter matrix of the convolutional layer tends to be higher than that of fully connected layers.

To investigate this, we designed a residual network with mixed layers, consisting of residual blocks of convolutional layers followed by residual blocks of fully connected layers. The architecture begins with an initial convolutional layer with $C = 256$ channels, a kernel size of 3x3, stride 1, padding 1, and no bias, followed by batch normalization and an activation function. This is followed by a stack of

$k = 5$ convolutional residual blocks, each with $C = 256$ channels and consisting of two convolutional layers (kernel size 3x3, stride 1, padding 1, no bias), each followed by batch normalization and a ReLU activation, with skip connections for residual learning. The output is then flattened and passed through a fully connected layer that adjusts the dimensions (input: $64 \times 32 \times 32$, output: $C = 256$), followed by layer normalization and an activation function. Finally, $k = 5$ linear residual blocks refine the features. Each block contains two fully connected layers (input and output dimensions: $C = 256$), layer normalization, non-linear activation, and skip connections. A fully connected classification layer maps the features to the number of output classes. We note that the minimal dimension of the weight matrix for both the convolutional layers within the residual connections and the fully connected layers is $C = 256$, so their ranks are always bounded by $C = 256$.

We trained this architecture on CIFAR10 with $\mu = 0.01$ and varying values of $\lambda$ and $B$. In Figure 16, we plot the terminal averaged rank of the convolutional layers within the residual connections and the averaged rank of the fully connected layers at the end of training, along with the effective ranks of selected individual matrices.

As shown in the figure, the averaged effective rank of both the convolutional layers and the fully connected layers, as well as the effective ranks of individual layers, decrease as $\lambda$ increases. Although the smaller dimension of the weight matrices is $C = 256$, the effective rank of all layers is smaller than 256 for both convolutional and fully connected layers. Interestingly, the effective ranks of the convolutional layers tend to be higher than those of the fully connected layers which aligns with the fact that the rank bound is higher for the convolutional layers.

**Singular values.** In our previous experiments, we measured the average rank of the weight matrices across different layers. To further investigate the rank of the learned weight matrices, we created visualizations displaying the singular values of the weight matrices for each layer as a function of batch size.

For instance, in Figures 18-19 we plotted the singular values of various layers for models that were trained in the setting of Figure 4(b) (main text) and Figure 13(c). Our results indicate that as a general tendency the singular values of each layer can be partitioned into two distinct groups: "small" singular values and "large" singular values (see the intersection point of all curves in the plots). Interestingly, the number of "small" singular values and "large" singular values is generally independent of the batch size. Moreover, "large" singular values decrease with the batch size and the "small" singular values increase with the batch size. This behavior provides additional evidence that when training with smaller batch sizes, the matrices have fewer large singular values compared to training with larger batch sizes.

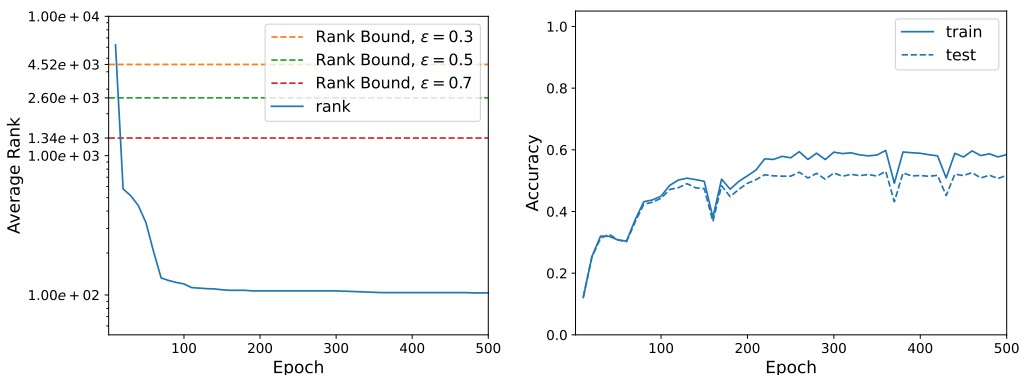

Figure 6: **Comparing our bound with the averaged rank.** We trained a MLP-BN-2-10000 on CIFAR10 with $B = 6$, $\mu = 0.1$, $\lambda = 8\mathrm{e}{-3}$. We plot our bound for different choices of $\epsilon$.

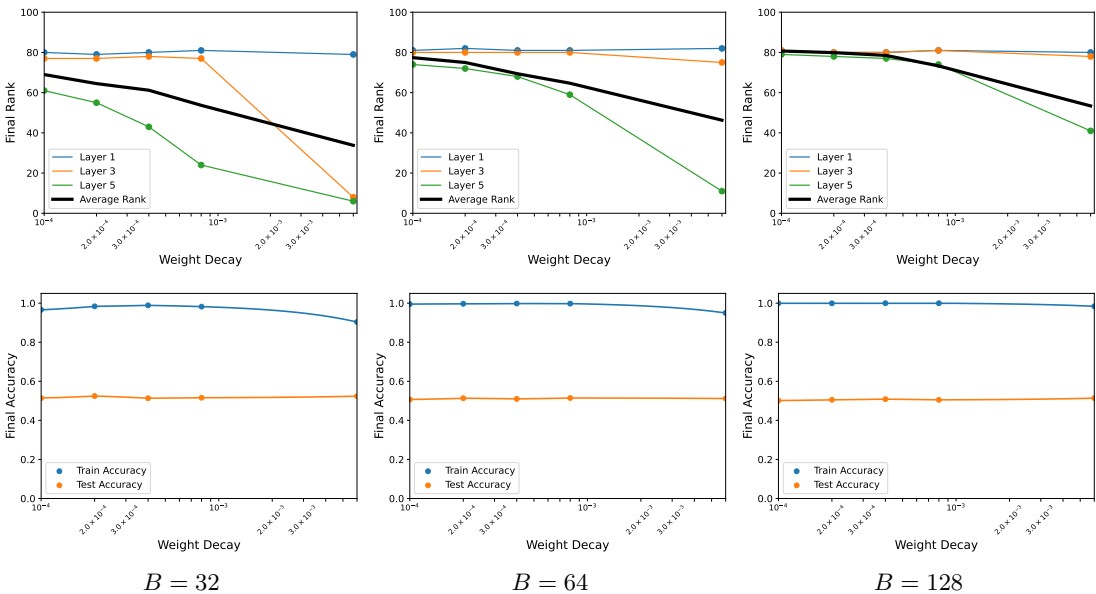

Figure 7: **Average ranks and accuracy rates of MLP-BN-10-100 trained on CIFAR10 when varying $\lambda$.** In this experiment, $\mu = 0.1$, momentum $0.9$ and $\epsilon = 1\mathrm{e}{-3}$.

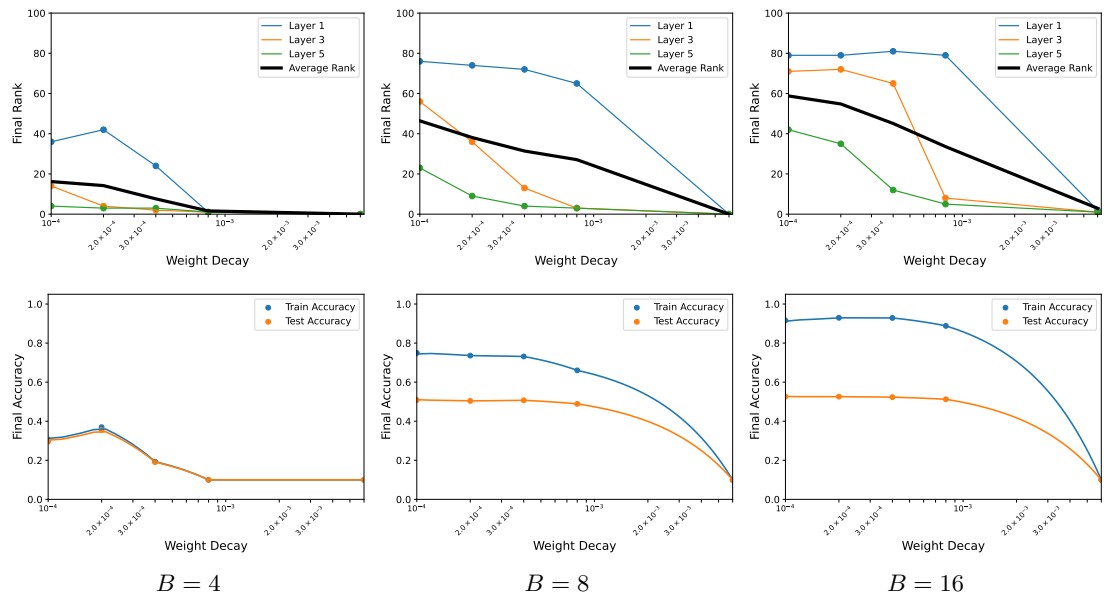

Figure 8: **Average ranks and accuracy rates of MLP-BN-10-100 trained on CIFAR10 when varying $\lambda$.** In this experiment, $\mu = 0.1$ and $\epsilon = 1e-3$.

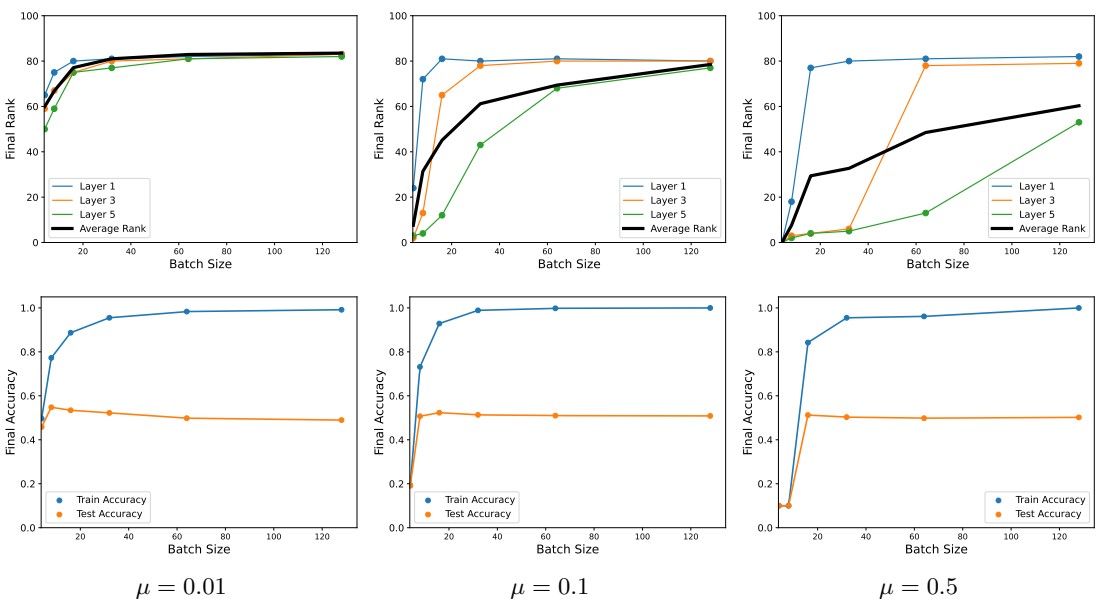

Figure 9: **Average ranks and accuracy rates of MLP-BN-10-100 trained on CIFAR10 when varying $B$.** In this experiment, $\lambda = 5e-4$ and $\epsilon = 1e-3$.

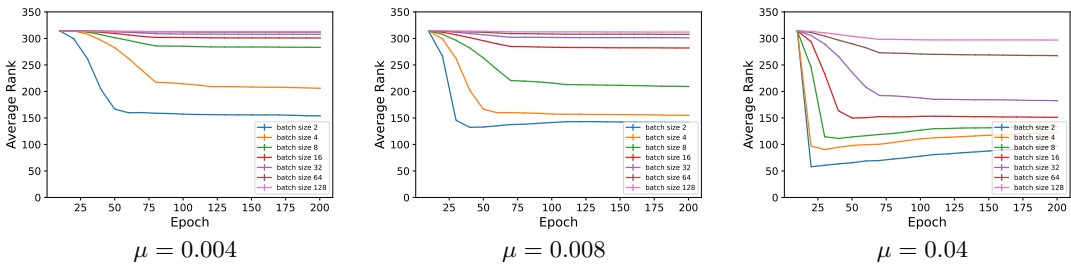

Figure 10: **Average ranks of ViT trained on CIFAR10 when varying** $B$. In this experiment, $\lambda = 5\mathrm{e}{-4}$ and $\epsilon = 1\mathrm{e}{-3}$.

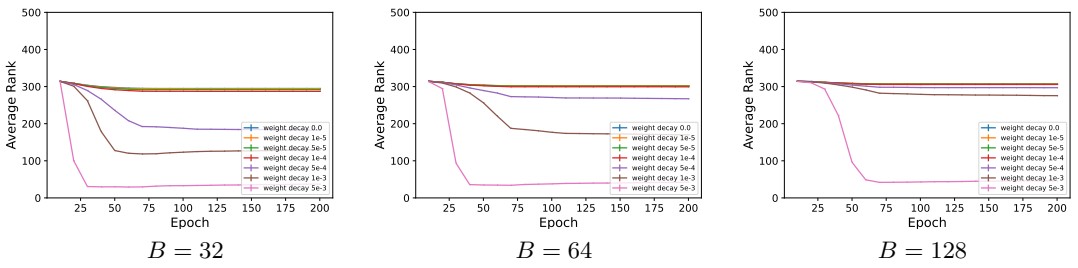

Figure 11: **Average ranks of ViT trained on CIFAR10 when varying** $\lambda$. In this experiment, $\mu = 4\mathrm{e}{-2}$ and $\epsilon = 1\mathrm{e}{-3}$.

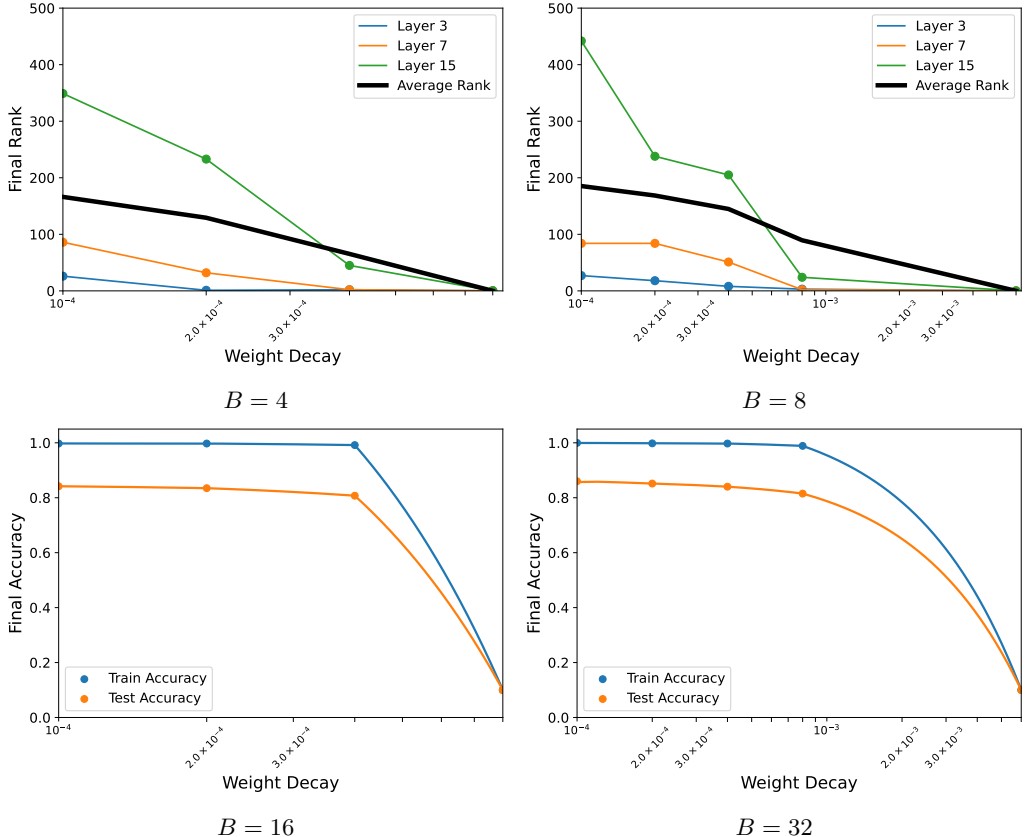

Figure 12: **Average ranks and accuracy rates of ResNet-18 trained on CIFAR10 when varying** $\lambda$. In this experiment, $\mu = 1.5$ and $\epsilon = 1\mathrm{e}{-3}$.

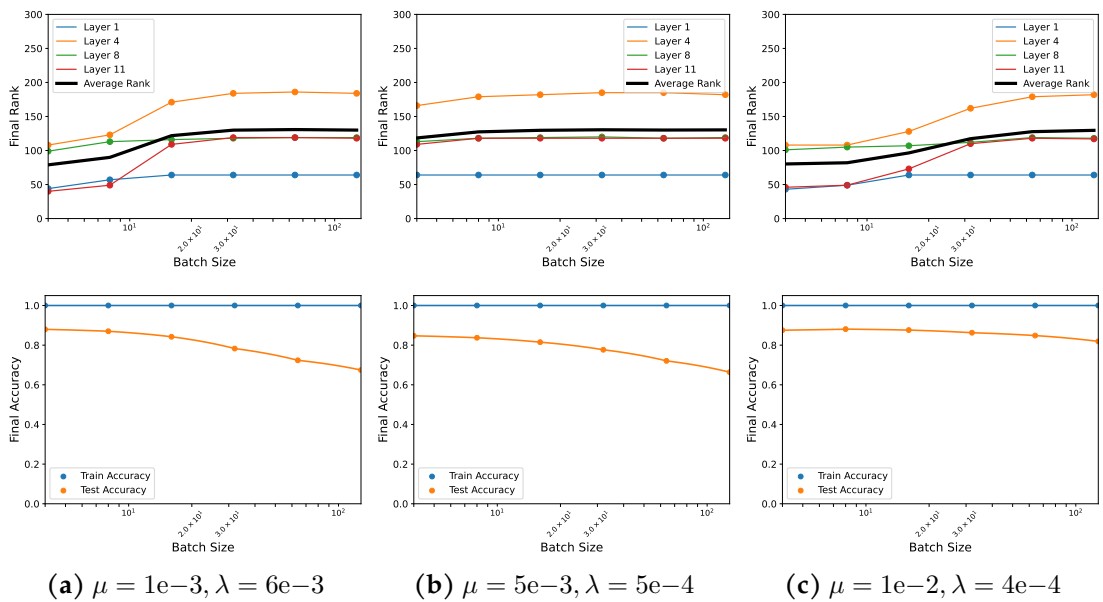

Figure 13: **Average ranks and accuracy rates of VGG-16 trained on CIFAR10 when varying** $B$**.** We used a threshold of $\epsilon = 1e{-}3$.

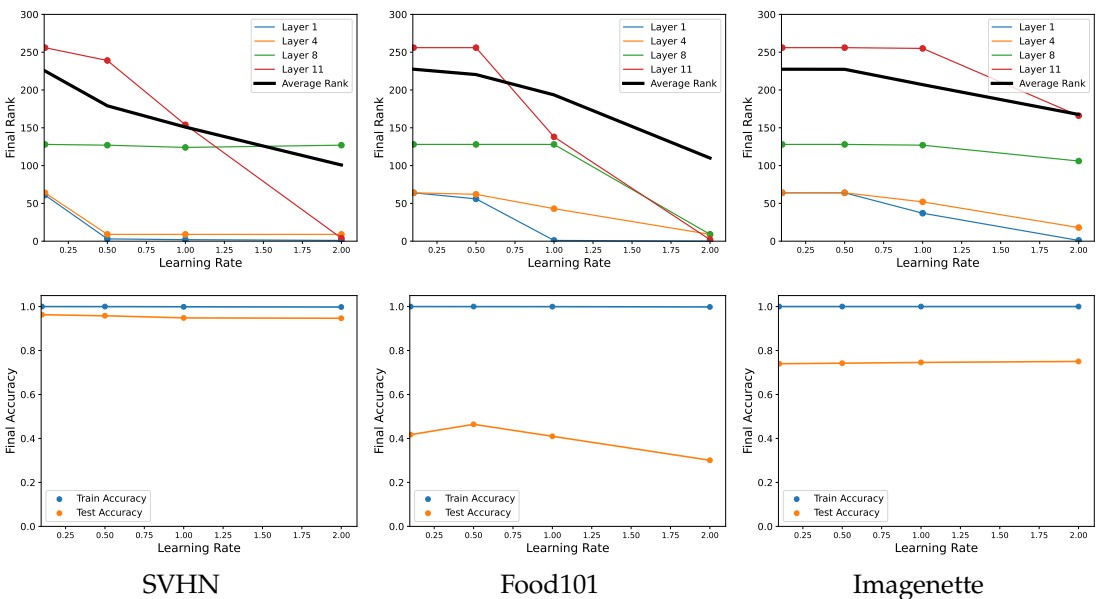

Figure 14: **Average ranks and accuracy rates of ResNet-18 trained on SVHN, Food101, and SVHN when varying** $\mu$**.** In this experiment, $B = 16$, $\lambda = 5e{-}4$ and $\epsilon = 1e{-}3$.

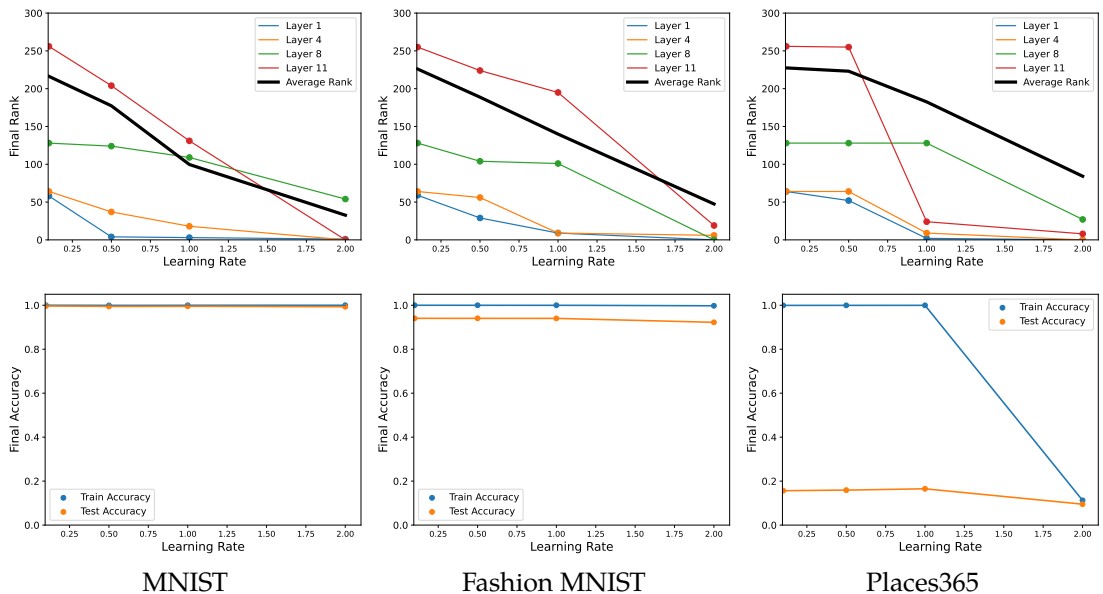

Figure 15: **Average ranks and accuracy rates of ResNet-18 trained on MNIST, Fashion MNIST and Places365 when varying** $\mu$**.** In this experiment, $B = 16$, $\lambda = 5e{-}4$ and $\epsilon = 1e{-}3$.

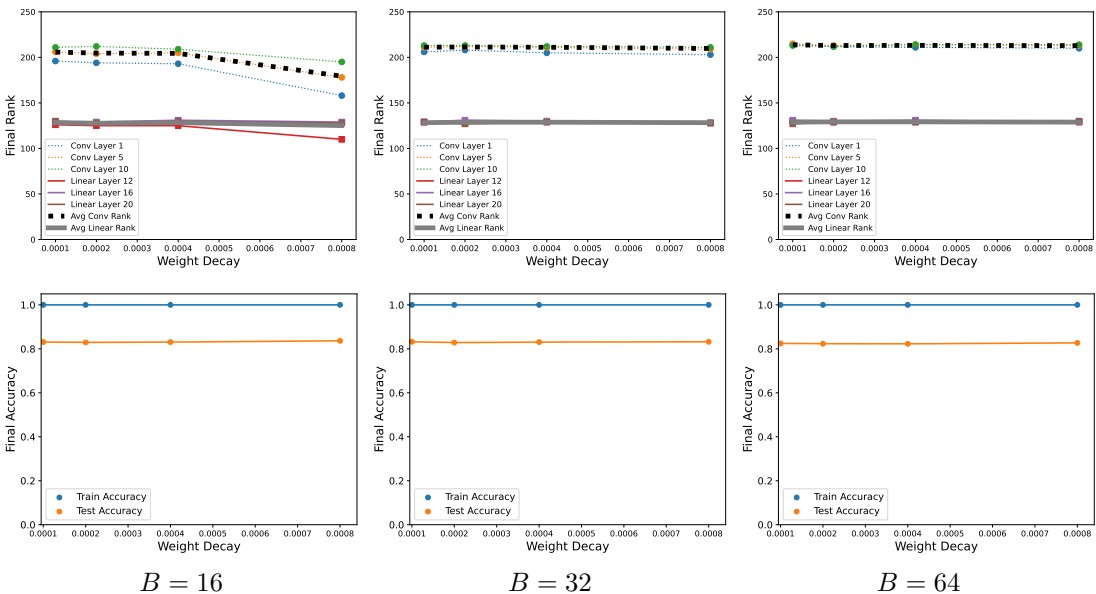

Figure 16: **Average ranks, individual layers' ranks and accuracy rates of a mixed residual network trained on CIFAR10 when varying** $\lambda$**.** In this experiment, $\mu = 0.01$ and $\epsilon = 1e{-}3$.

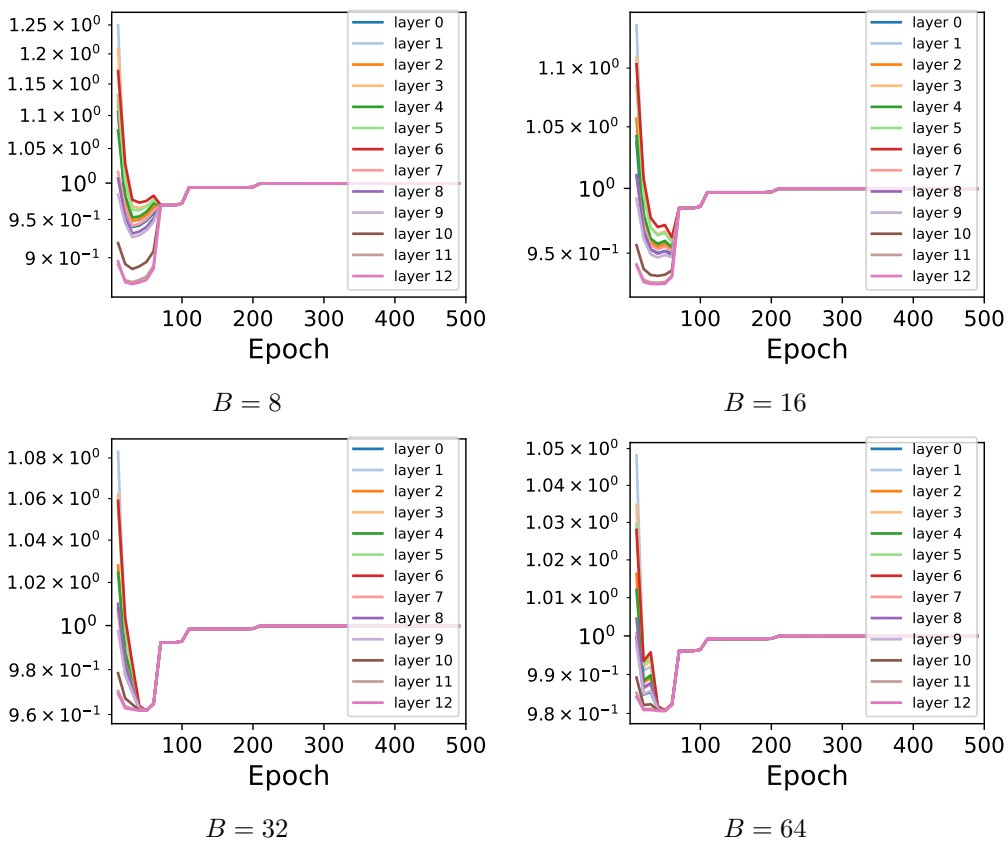

Figure 17: **Convergence of the weights for VGG-16 trained on CIFAR10.** In this experiment, $\mu = 5e-3$, $\lambda = 5e-4$ and $\epsilon = 0.01$ (see Figure 13(b) for the weight ranks and accuracy rates).

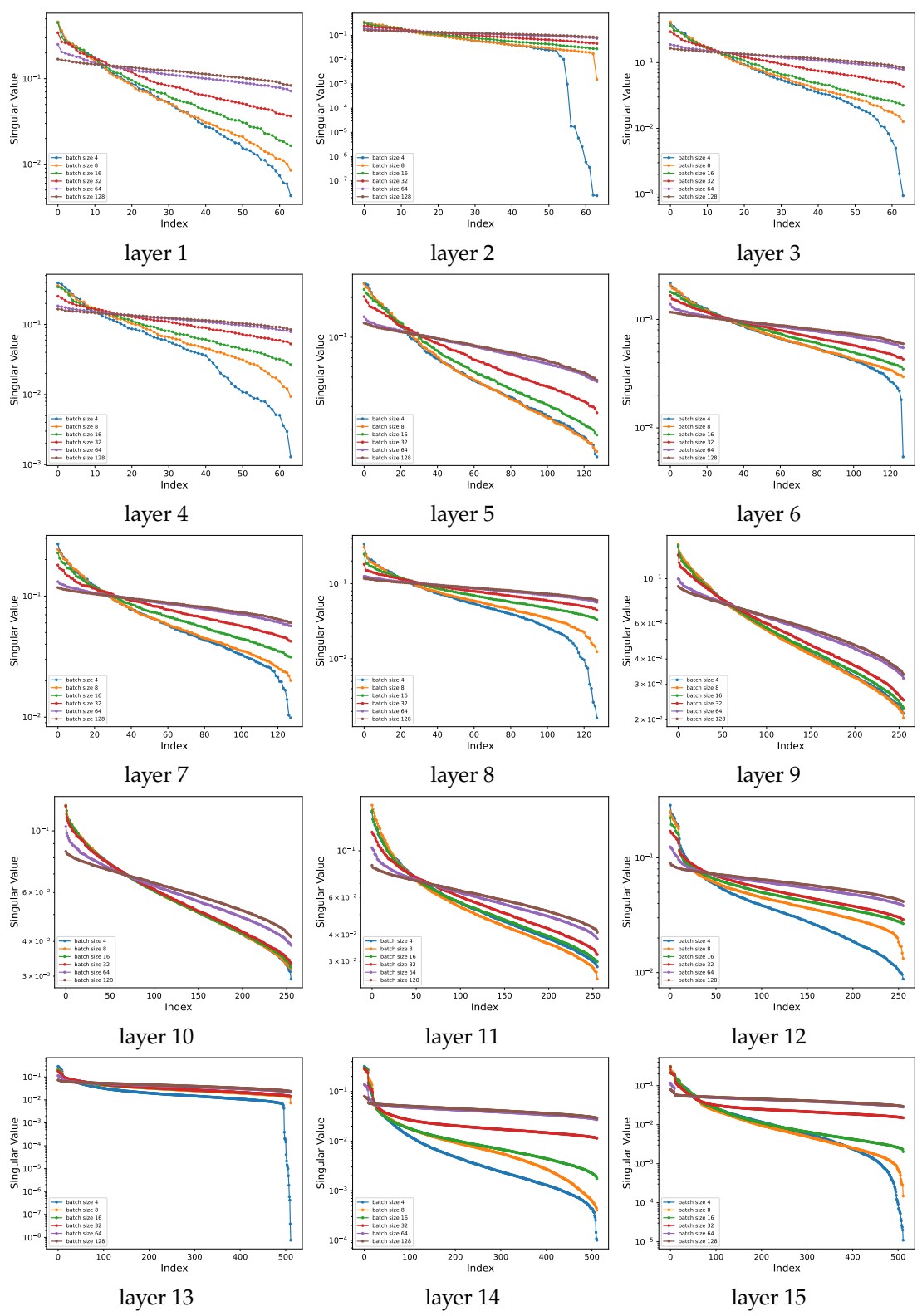

Figure 18: **Singular values of the weight matrices of ResNet-18 trained on CIFAR10 when varying** $B$**.** Each model was trained with $\mu = 5\mathrm{e}{-3}$ and $\lambda = 6\mathrm{e}{-3}$. Each plot reports the singular values of a given layer (see Figure 4(b) in the main text for the averaged ranks and accuracy rates).

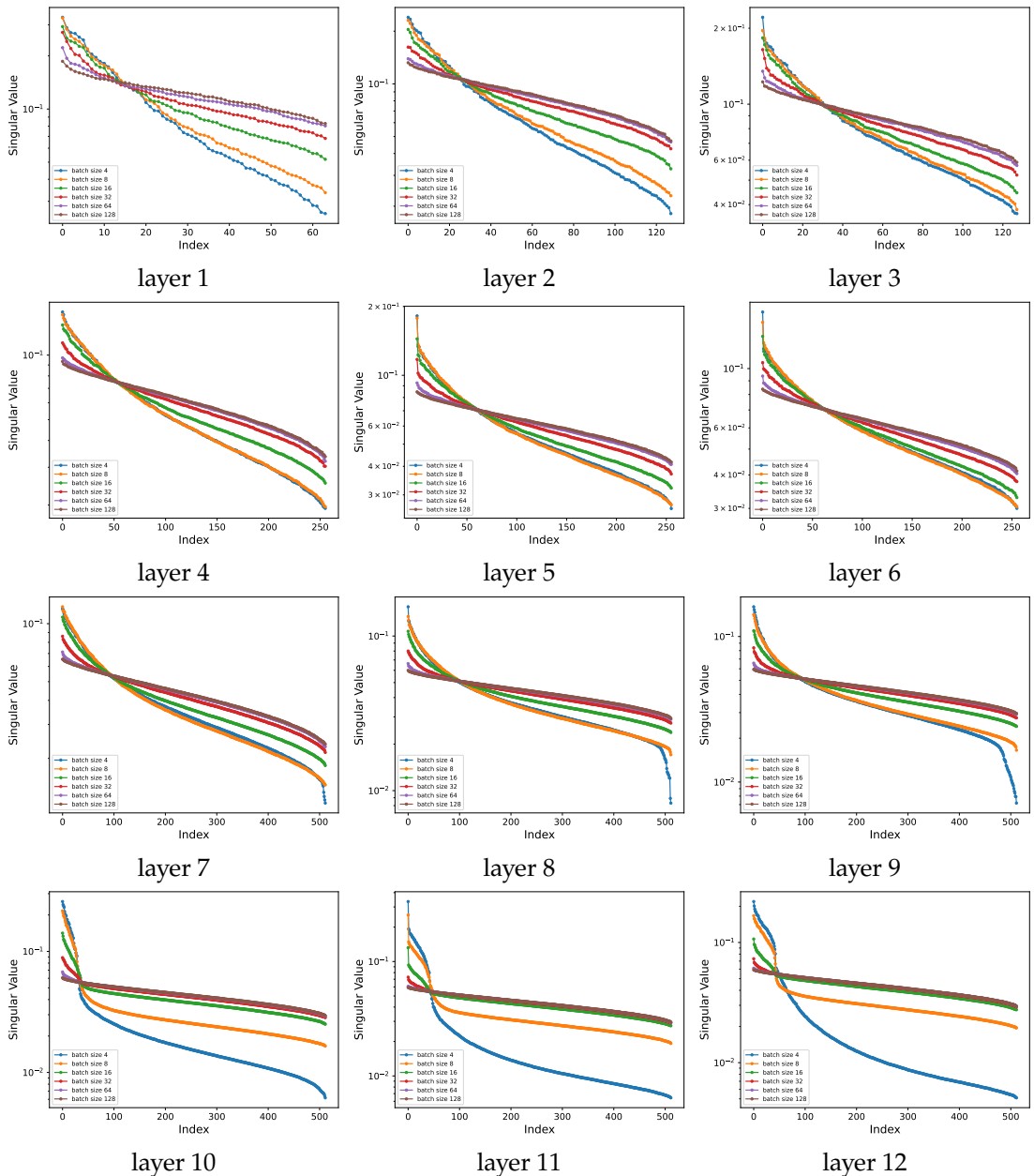

Figure 19: **Singular values of the weight matrices of VGG-16 trained on CIFAR10 when varying** $B$. Each model was trained with $\mu = 1\mathrm{e}{-}2$ and $\lambda = 4\mathrm{e}{-}4$. Each plot reports the singular values of a given layer (see Figure 13(c) for the averaged ranks and accuracy rates).

