# OpenReview forum: "SGD with Weight Decay Secretly Minimizes the Ranks of Your Neural Networks"
_CPAL.cc/2025/Proceedings_Track — CPAL 2025 (Proceedings Track) Poster_

### Official Review · Reviewer_P5Mq · 2025-01-08

**Rating:** 4
**Confidence:** 4

**Review:**

**Summary**: This paper studied the implicit bias of stochastic gradient descent toward learning low-rank matrices. The authors proved a bound on the rank of the weight matrices and showed that the weight matrices will have small rank under small batch size, high learning rate and strong weight decay.

**Clarity**: This work is clearly presented.

**Evaluation**: Although Theorem 3.3 presented the bound on the rank of the weight matrices after training which does not rely on assumptions on data, optimality of the learned weight matrices, etc, and the bounded rank is independent of training time $T$, I don't think this is very hard to show for **regularized** objective as in this case the optimal solution is bounded and thus the rank will be bounded. (Thus, the rank of the weight matrices after training will be independent of training time) Further, Theorem 3.3 actually relies on how fast the ratio $||W_{T-1}^l|| / ||W_{T}^l||$ approaches $1$ since the proof of Theorem 3.3 needs this ratio to be less than $2$.

For the experiment results, I don't think it is a good idea to plot the average rank of all the weight matrices across layers as the role of different layers may be different. I will just plot the rank of each weight matrices in each layer.

Thus, I will not recommend acceptance of this work.

---

### Official Review · Reviewer_DfLu · 2025-01-09
**Review of Submission47**

**Rating:** 7
**Confidence:** 5

**Review:**

This paper provides theoretical and experimental evidence that weights of various deep architectures trained by SGD are well-approximated by low-rank matrices, particularly with large weight decay, small batch size, and larger learning rates.

Pros
Theoretical results are general and well-presented, and reveal the effects of various hyperparameters, which are reflected in the experimental data.
Experimental validation of assumptions for the theorems is included.
Experiments are conducted on a wide variety of architectures, strengthening the claims made.
Cons
It is somewhat unclear how non vacuous the bounds from the theoretical results are, but this is likely an artifact of the general set-up.

As an aside, I believe the authors should cite the following paper, which theoretically connects low-rank bias to GD w/ weight decay for deep linear networks: Yaras et al. Compressible Dynamics in Deep Overparameterized Low-Rank Learning & Adaptation. ICML 2024.

---

### Official Review · Reviewer_6UzF · 2025-01-09
**Clean approach, but the presentation and implications need improvement.**

**Rating:** 7
**Confidence:** 3

**Review:**

### Summary:
This paper introduces a layer-wise analysis of the low-rank learning dynamics of weight matrices in neural networks. The authors aim to address one of the factors that induce low-rank bias in networks, which is the learning hyperparameters, using minimal assumptions.
The authors analyze the gradient with a straightforward intuition: for a linear layer, by the chain rule, the gradient of its weight matrix is the product of the layer's output and input, which is rank-1 for a single data point. (The authors also provide a patch-based extension for convolutional layers.) Under batch gradient descent, the rank of the batch gradient is therefore bounded by the batch size.
Using this bound, the authors sum over gradient updates to derive a final bound determined by the learning rate, weight decay, and batch size. They validate this by altering these hyperparameters and visualizing the resulting effective rank of the trained weights.

### Strengths:
1) Clever unrolling of the network structure, which encompasses fully connected (FC) layers, convolutional (conv) layers, skip connections, and other neural architectures.
2) By making minimal assumptions, the authors successfully isolate the impact of hyperparameters, revealing the relationship between low-rank bias, training hyperparameters, and model architecture. The bounding technique can also be extended by incorporating additional assumptions about loss functions or data for tighter bounds.

### Weaknesses:
1) Limited experimental scope: The experiments are restricted to relatively small datasets (CIFAR-10, SVHN).
2) The relationship between low-rank bias and generalization appears oversimplified. For instance, while a larger learning rate leads to a lower rank, it also results in worse generalization. Training parameters influence low-rank bias and generalizability, but the connection between rank and generalization remains unclear.
3) Figures 3, 4, and 5 could be better formatted. For example, a table or a summary curve showing the final rank/accuracy might be more effective. Plotting the full training curves seems unnecessary.

### Questions:
1) For FC layers, the structural constant $m_l$ seems much smaller than that for conv layers. In a hybrid network with FC and conv layers, does this imply that the rank of FC weights would be significantly smaller than that of the conv layer?
2) Regarding the universality of the unfolding approach: how would you compute $m_l$ for self-attention mechanisms?

---

### Meta-Review · Area_Chair_E3yS · 2025-02-06

**Recommendation:** Accept (Poster)
**Confidence:** 4

**Metareview:**

This paper provides a theoretical analysis of the implicit low-rank bias in neural network training, establishing a connection between training hyperparameters and the rank of weight matrices. The reviewers appreciate the clarity of the writing, the generality of the theoretical results, and the strong experimental validation across various architectures. However, one reviewer raises concerns about the novelty of the theoretical bounds and suggests that certain results may be straightforward under regularized objectives. I believe that the authors have addressed these concerns well.

Given the solid theoretical insights and broad experimental support, I recommend accepting this paper.

---

### Decision · Program_Chairs · 2025-02-11

Accept (Poster)